# Disjoint Spanning Tree Based Reliability Evaluation of Wireless Sensor Network

**DOI:** 10.3390/s20113071

**Published:** 2020-05-29

**Authors:** Sonam Lata, Shabana Mehfuz, Shabana Urooj, Asmaa Ali, Nidal Nasser

**Affiliations:** 1Department of Electrical Engineering, Jamia Millia Islamia, New Delhi 110025, India; smehfuz@jmi.ac.in; 2Department of Electrical Engineering, College of Engineering, Princess Nourah bint Abdulrahman University, Riyadh 84428, Saudi Arabia; 3School of Computing, Queen’s University, Kingston, ON K7L3N6, Canada; ali@cs.queensu.ca; 4College of Engineering, Alfaisal University, Riyadh 11533, Saudi Arabia; nnasser@alfaisal.edu

**Keywords:** network reliability, spanning trees, sum-of-disjoint products, wireless sensor network (WSN)

## Abstract

Wireless sensor networks (WSNs) are becoming very common in numerous manufacturing industries; especially where it is difficult to connect a sensor to a sink. This is an evolving issue for researchers attempting to contribute to the proliferation of WSNs. Monitoring a WSN depends on the type of collective data the sensor nodes have acquired. It is necessary to quantify the performance of these networks with the help of network reliability measures to ensure the stable operation of WSNs. Reliability plays a key role in the efficacy of any large-scale application of WSNs. The communication reliability in a wireless sensor network is an influential parameter for enhancing network performance for secure, desirable, and successful communication. The reliability of WSNs must incorporate the design variables, coverage, lifetime, and connectivity into consideration; however, connectivity is the most important factor, especially in a harsh environment on a large scale. The proposed algorithm is a one-step approach, which starts with the recognition of a specific spanning tree only. It utilizes all other disjoint spanning trees, which are generated directly in a simple manner and consume less computation time and memory. A binary decision illustration is presented for the enumeration of K-coverage communication reliability. In this paper, the issue of computing minimum spanning trees was addressed and it is a pertinent method for further evaluating reliability for WSNs. This paper inspects the reliability of WSNs and proposes a method for evaluating the flow-oriented reliability of WSNs. Further, a modified approach for the sum-of-disjoint products to determine the reliability of WSN from the enumerated minimal spanning trees is proposed. The proposed algorithm when implemented for different sizes of WSNs demonstrates its applicability to WSNs of various scales. The proposed methodology is less complex and more efficient in terms of reliability.

## 1. Introduction

Wireless sensor networks (WSNs) are formed by sensor nodes scattered over a large target area, to record, observe, or monitor as per the requirement of an application. Every sensor node performs various functions [1]. Application fields of WSN are environmental monitoring, pollution, landslide detection, traffic control and tracking, ongoing health control, industrial automation, military operations, and agricultural precision [2]. Each of these applications needs a reliable network to work efficiently. Each sensor node in WSN senses the information from the area within its range and sends it to the required destination according to the need of the application so that the sink node can get the required information for further processing. It is possible to classify WSNs according to the nature of the functions of the node. The categories are hierarchical networks, static networks, and defined networks of operations. In hierarchical networks, a sensor sets the priority by its position in the network. Transmission nodes have lower precedence than fully functional nodes (sensing, organizing, encoding, and forwarding information). The management of the network is carried out hierarchically and is specified based on their roles. In static networks, nodes are positioned in strategic positions before launching the application. The goal is to provide better performance in data collection and processing. Each node’s location is established, and the entire network is partitioned into disjoint clusters. The sensor network is a linked graph, where “N” is the set of vertices (sensor nodes) and “E“ is the set of edges. Sinks are predefined and non-mobile. The behavior of the node is specified when the network starts operating. The application begins after an event has been observed with the nodes and then the nodes forward their information to the objective node or the sink.

The dynamic nature of WSNs is greater when compared with wired networks, as nodes fail more often due to insufficient battery power and harsh application climate. Two major factors affecting network reliability are connectivity and the capacity to manage the traffic between nodes. Based on network connectivity, the "reliability" of WSN can be characterized as the probability of an operating path to exist between the source node(s) and the sink node. The reliability of WSN for communication and the reliability of WSN for an infrastructure communication phase was previously studied by using hierarchical, clustered, and tree algorithms. There is an extensive amount of research in the field of wireless sensor network reliability and it is divided into two explicit categories, (1) connectivity-based WSN reliability [3,4,5,6] and (2) flow-based WSN reliability [7,8]. The internal architecture of a sensor node is shown in Figure 1.

The sensors allow a sensor node to participate in the network not only by generating its own traffic but also by transmitting the traffic of neighboring nodes to the sink node. Major challenges in ensuring WSNs reliability are reliable links and nodes, link and node failure, and energy efficiency. It is equally important that the communication between sensors be secure and effective before the WSN revolution can fully take place. Any network interruption or loss of transmitted data will undermine user trust in the system. Because of the progressive dependence of information and communication technology on wireless networks, network efficiency is becoming one of the primary indicators for efficient design, planning, and implementation of WSN. The degree to which WSN is capable of delivering the requisite services needs to be measured quantitatively by identifying proper observable quantities. These observable quantities are called the measures of network reliability. The typical problem of network reliability is to measure the likelihood of all set of nodes or k set of nodes sending or receiving data from each other for a given period of time under some environmental conditions.

The probability of success of data forwarding from source to destination is called network reliability. Plenty of research has been done on network reliability for WSNs. Existing research mostly used the methods that are based on calculating the minimal path, sets, cut sets, and factoring theory. In minimal paths/cut methods the reliability is evaluated by enumerating all minimal paths/cuts and summing up the probabilities of their disjoint forms. Path and cut are formed by combining the part of a network such as links and nodes so that the system is said to be up or down, if nodes and links are in working condition, or failed correspondingly. Most of the researchers used a cut-based approach over the path-sets for calculating the reliability because it has been observed that a network has the least number of cuts as compared to path sets. In the proposed algorithm, we used the spanning-tree topology for computing the reliability of WSN. In comparison to paths/cuts-based approaches, spanning tree methods have proven to be more efficient. In the proposed algorithm, we chose an initial spanning tree, and then with the help of the initial spanning tree, we calculated disjoint spanning trees, which makes the calculation easier than path set or cut set-based approaches. The WSN reliability accommodating spanning tree algorithm can be calculated by transforming the network in the form of a linear graph having nodes and links representing centers for computers and communication channels, respectively. In this paper, we propose a network reliability analysis for WSN. This is also known as the global reliability of a WSN. Each achievable path between n (n-1)/2 pairs can be enumerated by the perception of terminal reliability [7].

Many researchers are working on finding the reliability of WSN [8,9,10,11,12,13]. To summarize, we can say that the proposed approach is very simple and straightforward as the reliability is calculated by considering the initial spanning tree (IST). It has also reduced the generation of non-spanning trees and eliminated failed spanning trees to avoid redundancy.

The paper is divided into different sections, starting with the introduction followed by Section 2 that describes the related work. Section 3 describes the problem statement and the reliability model. Section 4 describes the terminology. Section 5 represents the methodology used along with the algorithm by elaborating an example of a rooted tree. Section 6 comprises of the results and comparison, followed by a conclusion. Section 7 presents the abbreviation and acronyms used in this script.

## 2. Related Work

The probability of any system performing its functions efficiently for specific conditions in a specific time is known as the reliability of that system. The probability of success of the system depends on knowledge of the working conditions of all the components in the system. Once this information is available, the reliability of the system can be calculated. In the proposed algorithm, the case of link failure was taken into consideration. To deal with this measure of reliability, the concepts of graph theory were utilized. Multiple approaches are used for multi-terminal measurement, for example, vertex cut set, maximal cut set, spanning trees, and so on. All of these approaches are used to find the reliability of complex systems in case of a source to multi-terminal measure. With the help of graphical modeling, we can enumerate the network reliability of any system, considering path-set and cut set methods, spanning tree methods, and a lot more. Using a minimal path set approach, we estimated all the successful paths present to form a Boolean expression, also called a structure function. There are different minimal paths for different types of reliability calculation. For the two-terminal reliability calculations, all successful paths from a base station to the sink node needs to be discovered. Further, a Boolean expression from all successful paths is formed. Similarly, we can enumerate the two-terminal reliability with the help of cut sets. The number of terms in the reliability Boolean expression would be 2n-1, for n cuts or paths. Various techniques are available in the literature to reduce the Boolean expression of path sets and cut sets. A graph model of probability has been used by AboEIFotoh and Iyengar [2] for calculating the reliability of networks and it has been proved that it is an NP-hard problem. Some special wireless networks are studied by [14]. According to Park and Siva Kumar, sink to a single sensor node (Unicast), sink to a group of sensors (multicast), and sink to all sensors (broadcast) are three categories of models for data delivery. In 1981, Satyanarayan and Hagstorm calculated the source to multi-terminal reliability using a t-graph [15]. In this approach, after enumerating spanning trees they calculated the reliability in the factored form. The method generates all minimal spanning trees (non-disjoint) and makes use of a domination approach to assign the sign to the parts of reliability expression. Aggarwal and Rai extended a method by using a cut set approach. It also generates minimal spanning trees using cut set vertex approach and then by utilizing the exclusive operator generated all the disjoint terms [16]. Based on the same approach Feng and Chan gave a method of the spanning tree generator for source to all terminal reliability. It begins with a maximal cut set and then generates all disjoint spanning trees. To enumerate global reliability, Jain and Gopal suggested a method to find all the disjoint spanning trees [17]. The spanning-tree approach is one of the approaches that not only reduces the number of terms but also improves reliability [18].

For the health monitoring application of WSN, reliability can be calculated in three steps with the help of transforming the network in a propositional directed acyclic graph (PDAG) [19]. The binary decision diagram (BDD) is one of the efficient ways of reducing the size of the structure function for calculating the terminal reliability of the links. Common cause failure (CCF) is used in another algorithm [20] for calculating the reliability of WSN by utilizing an ordered binary decision diagram (OBDD). A clear procedure to compute k-coverage and k-connectivity, reliability by multiplication of all edges and vertices present in the network has been covered by [21,22]. Various reliability parameters of WSN reported in the literature have been studied viz. packet delivery ratio [23] and message delay [24,25]. Parameters such as link and node failure rates, environmental parameters such as noise, pressure, and temperature in [26] have been considered for developing efficient reliability of WSN. To obtain the network reliability of the overall system for communication purposes, it is essential to develop a network having a smaller number of links. The proposed algorithm proves to be a productive solution by using spanning trees.

Application communication and infrastructure communication are the two categories of WSN communication. Infrastructure communication works for transmission of the required information from the destination node to all the sensor nodes and application communication transmit observed data from all the sensors to the sink. The infrastructure phase keeps the network in a functional state for which it requires that the phase should have knowledge of present arrangement status for ensuring robust operation even in a hostile environment [27,28]. In this paper, a reliable measure based on connectivity for only static sensor networks has been presented, which is an initial step of infrastructure communication required to set up the network.

## 3. Problem Statement

This study anticipates the problem of modeling by enumerating disjoint spanning trees along with the reduced generation of failed spanning trees and assessing the reliability of WSNs directed to connectivity. We define the reliability of the WSN as the probability that a contact path exists between the sink node and at least one operational sensor in a target cluster. WSN’s reliability evaluation faces a range of problems that are not familiar with conventional networks due to special features of WSNs. Within the literature, there are numerous methods which have attempted to approach reliability evaluation problems by including accurate analytical estimation, lower and upper bound design, and simulation.

### 3.1. Assumptions

To carry out detailed theoretical reliability analysis of the WSNs to render the question of computing connectivity by using a probabilistic approach, the proposed analysis includes a few major assumptions. Nonetheless, for realistic WSNs which are used to acquire sensory data from the external environment, node failure, or environmental change can cause topology switching and therefore change the rate of data acquisition for each node. This paper focuses on the development of a route for reliability analysis of WSN between the source and sink nodes. This research explores the two-terminal reliability analysis of WSN from the network and will support WSN topology design, which includes the following assumptions:The first assumption is linked to edge failures being statistically independent. It is presumed that edge failures are statistically independent, which means the likelihood of a connection becoming operational is not contingent on the state of the other network links. The underlying assumption here is that communication failures are triggered by random events that individually affect all the connections. A static WSN consisting of a set of N nodes has been considered. While modeling the WSN into a probabilistic graph model for reliability calculation, it is usually assumed that all edge failures are statistically independent. The modeling of dependent link failures generally requires an exponentially large number of conditional probability distributions. Therefore, the independent assumption greatly simplifies the analysis of network performance.The second assumption is that the nodes are in operating condition. Any operable node shall remain operable for the entire duration of contact; therefore the model reflects a fairly short time as compared to the mean time between node failures (MTBF).The sensor node consists of four components, namely a sensor unit, a battery power unit, a micro-controller, and a transceiver unit. In this work, it is assumed that the power unit and the micro-controller are always operational. Mean-time-to-failure (MTTF) is relatively large as compared to the time required for transmission of messages average delay in propagation and the time required by the network to self-configure towards changes in topology due to failures. We also assume that the mean time to repair (MTTR) interval is fairly long. Thus, the state of the network is determined uniquely by the set of operational nodes during the message transmission time.All the nodes and links present in WSN are static. This means that the nodes of the sensor have no mobility. Data from sensor nodes are transmitted only to sink nodes that are final destinations. Transmission and reception between sensor nodes are not taken into account. Sensor nodes simply relay data to sink nodes. The positions were set for sensor nodes or sink nodes. The optimal position of sink nodes is not accounted for.Sensor nodes have a fixed range of communication.Communication channels or links are assumed to be half or full-duplex.The channel for communication between the links of wireless networks is perfect without having any barriers.Every link has two states, which are success and failure.The reliability of every possible link is known.

In this work, it is assumed that the nodes are reliable for the ease of testing them; however, in the future, the evaluation of WSN reliability with imperfect nodes as well as imperfect links can be examined. These assumptions are taken for mathematical convenience in calculations only and the proposed approach can be easily be extended for overriding these assumptions in the future.

We have stated that the operating probability of all links is 0.9 for the ease of calculations. The reliability of sensors is closely related to the sensing subsystem, the processing subsystem, the communication subsystem, and the WSN power supply subsystem. Modeling and testing the efficiency of the sensor node will be taken up as future work.

### 3.2. Reliability Model

There are three types of traditional network reliability:Source to terminal or two-terminal: In a network where there is a single source that generates information and a single terminal or destination where the information is desired to reach.Source to K terminal: In a network where there is a single information generating source and a set of K desired destinations, where K is an integer.Source to all terminals: In a network where there is a single source and the rest of the nodes are considered as terminals where the information is desired to reach.

One significant network reliability index is two-terminal reliability (2TR), which is the probability of a network’s two terminals (source and destination nodes) communicating successfully. 2TR evaluation is normally considered as the availability of a connection (path) from the source to the destination node, and is defined as the probability that there is at least one operational path between the source node and the destination node. This paper describes the availability of a network using two-terminal reliability, which represents the reliability of communication between a pair of nodes in a network. Although the proposed approach can also be modified further to calculate the all- terminal and K-terminal reliability. 

A WSN is represented by a probability graph *G = (n, r)*, where *n* is the set of sensor nodes and *r* is the set of network connection links. There is a contact link between two nodes when they are within each other’s radio transmission range. It can be observed from Figure 1 that each sensor node consists of four components: a sensor unit, a battery power unit, a micro-controller, and a transceiver unit. Sensor nodes are prone to unpredictable failures due to the harsh existence of the application world at WSN. Either battery unit or micro-controller failure leads to full node failure. Therefore, the battery unit and the micro-controller are always considered active in this study.

For building a WSN model the graph of a WSN has to be created. For this analysis, it is presumed that all the sensors belonging to a WSN are identical. If a sensor called A is within a sensor B communication range, then sensor B is also within sensor A’s communication range, and in this case, AboElFotoh’s method [2] of obtaining an undirected graph for a WSN model was chosen.

The basic concept behind this approach is depicted in Figure 2, where a wireless network and its corresponding graphical model can be seen. The diagram shows that there is a bidirectional edge between each node if they are within the range of each other. The edge A–B in Figure 2 implies that there is a contact path between A and B and between B and A. We use this edge in this study to represent the bidirectional edge. Following are the details for the WSN graph network (*N*) that we have used for the analysis:
*N* = (*G*, *P*, *s*, *T*),
where
(1)*G* = (*n, r*) is a graph with the set of nodes *n = (n1, n2, …, nm*) and *r = (r1, r2, …, rk*),(2)*P* = {*p(r1), p(r2), p(rk)*} denotes the operating probability set of *p(rk)*, and 0 ≤ *p(rk)* ≤ 1,(3)*s* represents the source of the WSN,(4)*T* = (*t1,t2, …, tn*) represents the set of sinks in the WSN, where *n* is the number of the sink nodes.

## 4. Terminology

The reliability of any system plays a very important role in improving its performance. Reliability improvement is one of the most important requirements of WSN for its various applications, whether it is an industrial or medical application. Reliability modeling is one of the easiest and less time-consuming techniques for estimation of reliability. For the optimization of WSN, reliability modeling is an important parameter. Although this brings the WSN into effective action, reliability modeling helps in anticipating the nature and working of network components so that we can design WSN accordingly. For the WSN already in an effective mode, reliability modeling could help in finding the critical components and improving reliability. Existing non-simulation approaches are normally based on Boolean algebra, fault tree (FT), binary decision diagram (BDD), and Markov chains, etc. Binary decision diagrams (BDDs) are one way to analyze the reliability issues of complex systems that consist of many components as an appropriate method for representing them to allow the efficient storage of system topology information. The sum of the disjoint product (SDP) method for reliability evaluation has been used in this work.

### Sum of Disjoint Product (SDP) Technique

The Sum of the disjoint product as given in [29,30] uses the knowledge of a minimal path and minimal cuts of reliability graph. The minimal path and minimal cut are a set of minimal number of components whose working and failure affect the whole system. SDP exploits the addition law of probability, i.e., the probability of occurrence of one of the events with two or more events having no element in common will be the sum of probability of individual events. For example *P(A U B)* = *P(A) + P(A’B);* for n elements *P(A1UA2U.....An)* = *P(A1) + P(A1 ’A2) + P(A1 ’A2 ’A3)*......... *P(A1’....An−1’An*). In the sum of the disjoint product method, every minimal path or minimal cut is made disjointed with one another. The method of making every path disjoint starts with taking into consideration the present term and its predecessor’s term. The element, which is found in common between the two terms, is deleted from the predecessor’s term. Union of the two terms is taken and then the complement of the union is taken. The expression is solved by applying Boolean algebra and hence made disjoint with one another. The same procedure is followed by third, fourth, and up to n number of terms. In this way, every path is made disjoint and no repetition of paths occurs during the application of the method.

## 5. Methodology

A graph is a symbolic way of describing any network, and therefore, algorithms of network analysis can be easily implemented to study the various reliability-related performance indices like tie sets, spanning trees, etc. The subset of a graph is called a subgraph and a tree is a formed by a circuit less graph. *G (n, r)* is a graph with ‘*n*’ specifying the nodes and ‘*r*’ describing the edges/links. The spanning tree of the graph *G* has all the nodes of *G* with no cycles.

The IST is a specific spanning tree, which is formed by retaining the node with the maximum number of edges incident on it. If a network has two nodes with a maximum or same degree of incidence, any one of them can be retained to form IST. A rooted tree is a tree that has an initial component called root to start with. In this paper, IST acts as the root of the tree. Failed IST or IST’s are a tree formed by the rest of the edges that are not included in the IST. Disjoint spanning trees are formed by appending edges of the IST’ when the edges of the IST fails. The union of disjoint spanning trees results in the formation of the successful spanning tree. During the tree-generation process, no duplicate spanning trees are created, which is why they are called disjoint spanning trees. ST is a spanning tree that is formed by the union of disjoint spanning trees.

If the source node can link with the sink node, then it would become a connected graph. A fully connected network has all the pairs of nodes connected either directly or with the help of other sensor nodes. For connectivity in WSN, we need to have a reliable number of paths between sensor nodes. Parameters affected by connectivity in the graph are robustness and throughput of the wireless sensor network. For enhancing reliability in WSN, coverage and connectivity are the principal target parameters.

For any network, the algorithm first begins by utilizing a specific spanning tree; also known as the initial spanning tree (IST). IST has a node on which the maximum number of edges are incident and it has one lesser node than the total number of nodes present in the graph i.e., (l = n−1). A set *S* is formed by the edges of IST and the rest of the edges are moved to set *T*. Main steps for generating disjoint spanning trees are as follows:If one edge fails from set *S*, then every edge from set T will get appended to IST by deleting the failed edge to check the probability of success of immediate successful spanning trees (STis). STi is the set of successful spanning tress obtained.If two edges fail at a time from a set *S*, then all the possible combinations of two edges at a time from *T* with IST will get appended by deleting the two failed edges to check the probability of success of STis.

Then repeat it by three edges at a time and continue in the same fashion. This procedure ends when IST consists of failed edges more than the elements present in set *T*. The procedure given in Rule 1 is followed to reduce the generation of unsuccessful/failed spanning trees when combinations of two/three …,*k*, failures in IST are considered.

### 5.1. Network Model and Preliminaries

**Rule 1:** “When the edge(s) connected to the node of incidence in the graph fails in a given IST then all those elements of *T* which do not contribute in STi formation, also fail to contribute. Thus, the formation of these failed trees can be avoided.”

For the network shown in Figure 3, when an edge *r3* in IST of Figure 4 is failing, then only *r4* contributes to STi formation. *r7, r8*, and *r9* have no contribution to STi formation. When *r3* has failed in combination with other failed edges from IST then the combination of *(r7, r8); (r7, r9); (r8, r9)* and *(r7, r8, r9)* will never contribute in STi formation.

**Rule 2:** “When failed edge(s) isolate a single node then only those edges that are incident on the isolated node in the graph will be able to yield STi when appended with the isolated node in IST’. In Figure 4 when *r1* fails, then node *n1* is isolated”.

**Rule 3:** “When failed edge(s) in the IST isolate multiple nodes at the same time, then the graph is divided into two subgraphs. Only those edges from set *T* are appended to IST’ which connects the nodes of isolated subgraphs”. 

The IST for the network shown in Figure 3 is represented in Figure 4. When edge *r1* is failed, then the graph gets separated into two sub graphs as shown in Figure 5. All the edges that connect the first subgraph to another subgraph are taken from set *T* and appended with IST’ to form STi as shown in Figure 5.

**Rule 4:** “When failed edge(s) isolate a single node and multiple nodes simultaneously, then at first those edges that are incident on the isolated node in the graph are selected from set *T*. Then the edges which connect the subgraph formed due to multiple edge isolation, to the rest of the subgraph is selected from set *T* and appended with IST’ to form STi.”

**Rule 5:** “A new set *C* is formed which contains all the circuits or closed contours present in the network. Whenever an edge is selected from set *T*, the set *C* is checked to avoid the generation of circuits in the network, which leads to the generation of failed spanning trees.”

For *G (6, 9)* in Figure 3, the elements of set *C* are: *{r1r6r7, r5r7r8, r2r8r9, r3r4r9, r2r3r4r8, r1r5r6r8, r2r5r7r9}*. When these rules are implemented with the algorithm, it leads to the following advantages:The formation of a failed spanning tree is reduced.Duplicate spanning trees are not generated.The algorithm might be slow, but the generation of disjoint spanning trees takes less amount of time.

### 5.2. Algorithm

The algorithm 1 was implemented using C language. The proposed algorithm contains two modules. The first module of the algorithm describes the initialization part, and the remainder of the module caters to network connectivity. It finally produces all spanning trees for terminal-reliability evaluation.

**Step1.** Settle on a spanning tree containing a node with a maximum incidence of edges on it. This consists of a minimum number of edges forming a connection between the source node and the terminal node. This spanning tree is called as IST. Apply rule 1 to avoid the formation of failed spanning trees and place the edges of IST in set *S*.
**Algorithm 1****Initial Spanning Tree****START**1. *G(N,E)* = graph with n number of nodes and r are the edges.IST = Initial Spanning TreeQ = Queue.Ns = Starting Node2. Ns = Select the node with maximum number of edges incident on it.3. Q.enqueue( Ns )mark Ns as visited.while (Q is not empty){IST = Q.dequeue(·)//processing all the neighbors of *v*for all neighbours w of *V* in Graph *G*{if w is not visited{Q.enqueue(·w·)mark was visited.}}}**END**

**Step2.** Place the rest of the edges of IST in set *T*.

**Step3.** (i) A link can fail in ^m^C_1_ possible ways in IST. There are ^k^C_1_ ways a link from *T* can be appended to each IST’ to check the probability of success of STi formation. All ST_i_ ’s so generated are retained. Apply rule 2.
1. **START**2. **Initialization of Variables**3. IST = Initial Spanning Tree;4. IST’ = Failed Initial Spanning Tree;5. *n* = Number of nodes;6. *r* = Number of edges;7. *G (n,r)* = Graph of ‘n’ nodes and ‘r’ edges;8. *S* = Set of edges in IST;9. *T* = Set of edges excluded from IST;10. *C* = Set of circuits (loop) in the graph *G(n,r)*;11. ST_i_ = Intermediate Spanning Tree;12. ST = Successful Spanning Tree;13. **Create an IST of *G(n,r)* from first algorithm**14. *S* = [set of edges in IST];15. *m* = number of edges in *S*;16. *T* = [set of edges excluded from IST];17. *k* = number of edges in *T*;18.   For (i = 1, i < = *k*, i++){   IST’ = IST – linkFailure(^m^C_i_ S[]);   ST_i_ = Apply Rule (IST’ + link Append(^k^C_i_
*T*[]);}19. **Remove the circuit loop to get the total successful spanning tree.**20. ST = U_i_ STi21. R = ∑P (STi)22. **END**

(ii) Two links at a time can fail in ^m^C_2_ possible ways in IST and ^k^C_2_ ways are possible for appending two links from *T* to each IST’ to check the probability of success of STi formation and apply rule 3. (iii) Repeat the procedure in (ii) until all the possible combinations of elements present in *T* with three and more or k failures at a time and then apply rule 3 and rule 4.

**Step4.** (i) There are edges in set *S*, which do not result in STi formation after the edges of set *T* are appended when failed (one at a time) in IST. The rest of the edges of the set *S*, in conjunction with these failed edges of set *T*, will also fail to produce STi. Therefore, the development of failed trees is evaded from these edges and their combination. This minimizes failed spanning tree formation. (ii) When failed edge(s) isolate a single node then only those edges which are incident on the isolated node in the graph will be able to yield ST_i_ when appended with the isolated node in IST. (iii) When failed edge(s) in IST isolate multiple nodes at the same time, then the graph is divided into two subgraphs. Only those edges from set *T* are appended to IST’ which connects the nodes of isolated subgraphs. (iv) When failed edge(s) isolate a single node and multiple nodes simultaneously, then at first those edges which are incident on an isolated node in the graph are selected from set *T*. Then the edges that connect the subgraph formed due to multiple edge isolation to the rest of the subgraph are selected from set *T* and appended with IST’ to form STi. (v) A new set *C* is formed which contains all the circuits or closed contours present in the network. Whenever an edge is selected from set *T*, the set *C* is analyzed to avoid the generation of circuits and the formation of failed spanning trees.

**Step5.** WSN reliability expression can be accomplished by taking the union of disjoint spanning trees and changing the logical variables of each spanning tree to corresponding variables of the probability of system success function (SSF).

### 5.3. Illustration

This paper proposes an algorithm to compute reliability. The method is simple and applies to the network having directed or undirected links. Consider the graph in Figure 6 i.e., *G (5, 7)*. The maximum incidence is on node *n3*. To form IST as shown in Figure 7, all edges connected to node *n3* are retained.

In this example *S* = {1, 2, 6, 7} *T* = {3, 4, 5} and *C* = {156, 237, 467, 1457, 2346}. First, the edge 1 from set *S* is failing, and we append edge from T with IST one at a time to form STi’s. Edge 3 and edge 4 fail to generate ST_i_ with 1267. Table 1 shows the formation of all the ST_i_ ’s formed by appending edges from *T*. In the same way, other edges one at a time, then two and so on from *S* have failed independently and elements of *T* are appended to check the probability of success of ST_i_’s formation. Edges with complement sign (‘) represent the failed edges.

The case of two edges failing at a time is depicted in Table 2. Edge 1 is combined with another failed edge and edges 3 and 4 together fail to contribute to ST_i_ formation. This stops the formation of all the failed spanning trees generated with edges 3 and 4. Likewise, when edge 2 is failing, then edges 4 and 5 have no contribution to STi formation. So when 2 is combined with another failed edge, then edge 4 and edge 5 fail together to contribute to STi formation. This stops the formation of all these failed trees with 4 and 5 as an appended edge.

Figure 8 shows a rooted tree for *G (5, 7)* which is obtained by applying the algorithm. The tree is a rooted tree with IST as the root. Failed edges of IST are represented by the sign of compliment in the rooted tree. (1’, 2’, 6’, 7’) are the complimented edges of the IST. Dark dots represent successful spanning trees and light dots represent failing trees. Dotted lines represent those links that are independent or when combined do not contribute in any ST_i_ formation. Failed spanning tree generation is stopped and is represented by the symbol “X” on the rooted tree. Overall, 21 successful spanning trees were obtained. The repetition of a successful spanning tree in consecutive steps was stopped. SSF is the union of all disjoint spanning trees. For obtaining the reliability expression, each spanning tree logical variable is multiplied by corresponding probability variables to obtain X_i_ p_i_ and X_j_q_j_. STi contains complimented and non-complimented mixed variables. Table 2 shows disjoint spanning trees.

System success function, S, may be written as: S = U_i_ ST

Reliability expression is obtained as: R = P(U_i_ ST_i_) = ∑P Ui (ST_i_)

The SSF, is obtained by the union of all spanning disjoint trees that have been created. The expression of reliability can be obtained by modifying:Union operator by summation operator in the function of SSF.Logical variables for each spanning tree to corresponding variables of probability ( p_i_ and q_j_).

The algorithm generates a rooted tree in which root vertex represents IST. The vertices of the rooted tree have been marked as a dark black circle and are shown in Figure 8.

## 6. Results and Discussions

In Figure 6, the Aggarwal method [17] generates the same number of terms in reliability expression as the proposed method, but the method of Aggarwal follows a two-step approach. Firstly spanning trees from fundamental cut sets are obtained and then made disjoint by applying a disjointing technique. Moreover, the criteria used for fundamental cut set selection are arduous. Formation and repetition of non-spanning trees occur after taking the Cartesian product of fundamental cut sets when cut sets are chosen randomly.

In Jain and Gopal method [18] successful disjoint spanning trees are generated, but they have not proposed any technique to reduce the generation of failed spanning trees. The proposed method has the capability of stopping the generation of failed spanning trees at different stages of the algorithm.

Examples of some network have been presented in Figure 6 and Figure 9. Table 3 represents the comparison of computational efficiency of different methods implememnted on these networks. It is also evident from these implementations that the number of disjoint spanning trees, thus formed by the proposed algorithm is always equal to or less than the number of disjoint spanning trees. This implies that the number of terms in reliability expression has been always less in number. Other disjointing methods generate disjoint spanning trees generally more than or at most equal to the minimum spanning trees.

We applied the proposed algorithm to other networks of varying complexity from literature and calculated their reliability. Experimental findings obtained from the published research on multiple networks are presented in Table 4. Among these, the comparison of spanning trees and path set with disjoint spanning trees for a few benchmark networks (shown in Figure 10, Figure 11, Figure 12, Figure 13, Figure 14 and Figure 15) are given in Table 4, which exhibits the efficiency of the algorithm and the proposed framework for evaluating global reliability using the connectivity criterion.

In addition, Table 4 also provides an enumeration period in microseconds. Reliability obtained by the proposed approach and by the inclusion–exclusion method has been compared for each network. Various benchmark networks (Chaturvedi and Misra 2002 [30]; Gebre and Ramirez-Marquez 2007 [31]; Chakraborty and Goyal 2015 [32]) available in the literature are presented in Figure 10, Figure 11, Figure 12, Figure 13, Figure 14 and Figure 15. Table 4 presents the performance comparison.

Table 4 shows the efficacy of the proposed algorithm for assessing reliability measures using connectivity criteria. Experimental findings obtained from the previous research on several networks indicate that the number of disjoint terms turns out to be either less or the same as the number of spanning trees.

## 7. Conclusions

Connectivity is an important aspect of network reliability measure for wireless sensor networks and has been studied by several researchers. A lot of research has been carried out for calculating WSN reliability, but most of the research is based on a cut set and a path set. Very few researchers have explored the spanning-tree approach, which is also a viable option. The proposed algorithm requires neither minimal paths/minimal cuts to be enumerated in advance. The reliability of WSN is evaluated with the help of disjoint spanning trees, which is generated by considering the case of link failure. The advantage of this method is that it requires only one initial spanning tree to begin with. The method yields the minimum amount of disjoint spanning trees. The proposed method is fast, efficient, and it generates only successful disjoint spanning trees and stops the formation of failed spanning trees. When the complexity of WSN increases, the proposed method is better in comparison to the path set and cut set approaches in terms of time consumption. The approach is described with the help of an example. Table 4 illustrate the computational complexity of the proposed algorithm. The proposed approach is conceptually simple and computationally effective, as it takes less computational time to list subset cuts for networks with edge failure as compared to other strategies present in the literature. The proposed method can be extended to consider delay and capacity parameters in the future by making further amendments to the proposed algorithm in this paper. The proposed solution can be extended easily for multi-source, multi-terminal networks with unreliable nodes and edges.

## Figures and Tables

**Figure 1 sensors-20-03071-f001:**
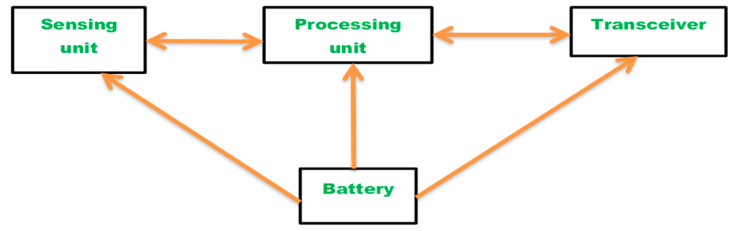
The inner architecture of a sensor node.

**Figure 2 sensors-20-03071-f002:**
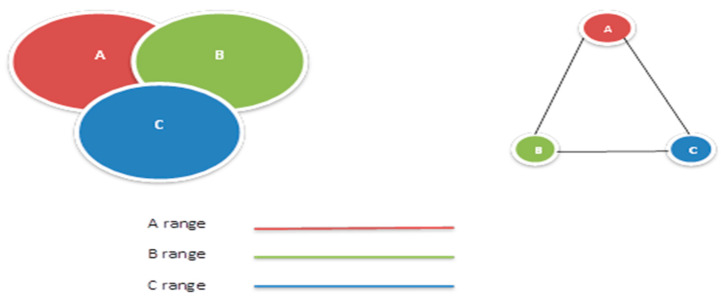
Wireless network and the corresponding graph model.

**Figure 3 sensors-20-03071-f003:**
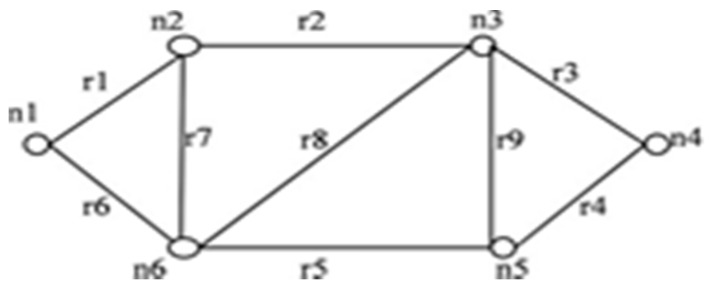
*G (6, 9)*.

**Figure 4 sensors-20-03071-f004:**
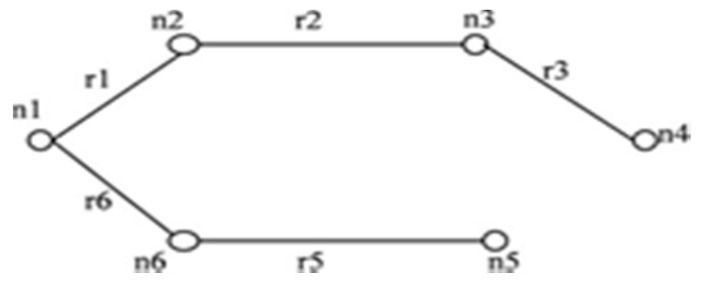
Initial spanning tree (IST) for *G (6, 9)*.

**Figure 5 sensors-20-03071-f005:**
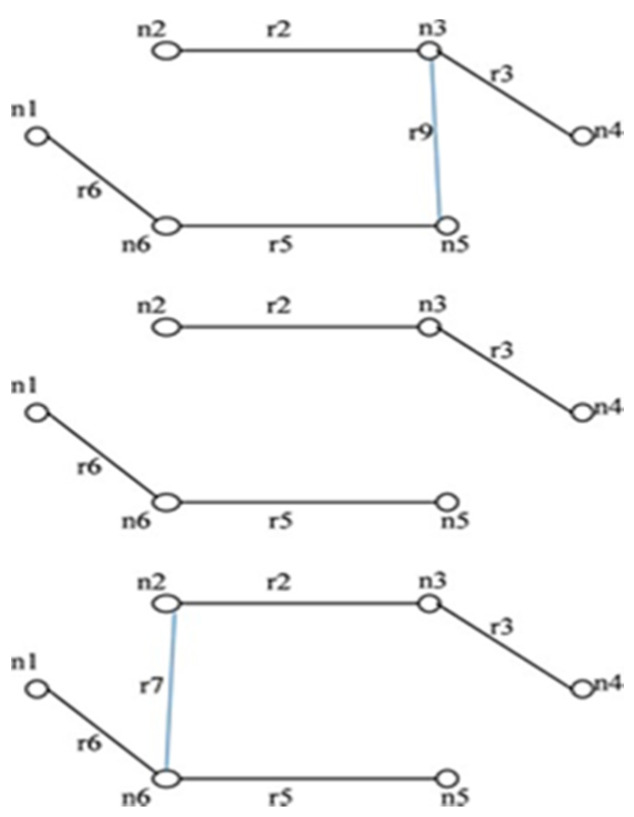
Illustration of rule 3.

**Figure 6 sensors-20-03071-f006:**
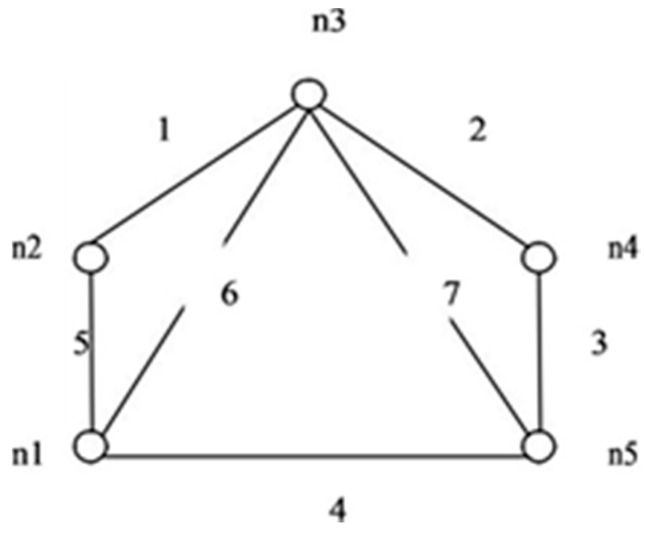
*G (5,7)*.

**Figure 7 sensors-20-03071-f007:**
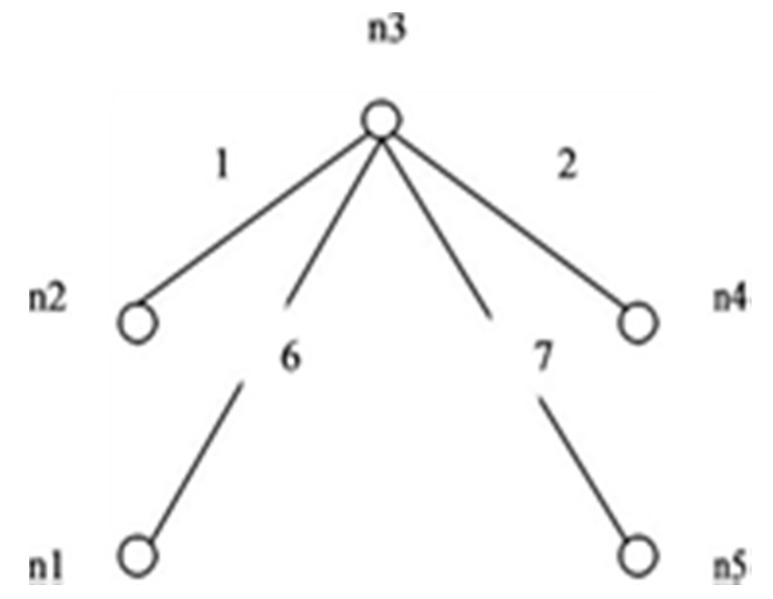
IST for *G (5, 7)*.

**Figure 8 sensors-20-03071-f008:**
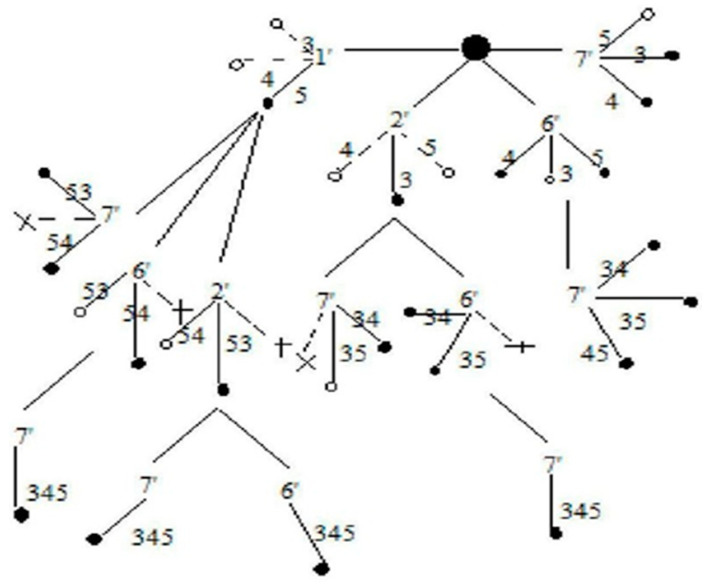
A rooted tree for *G (5, 7)*.

**Figure 9 sensors-20-03071-f009:**
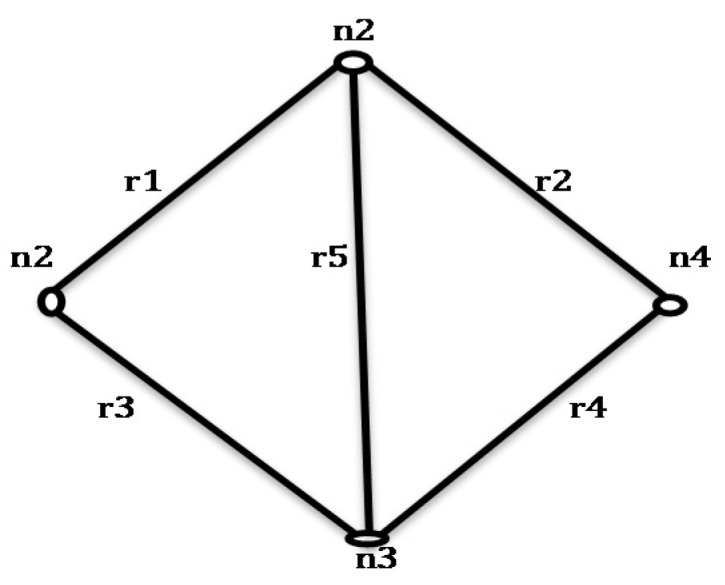
Bridge Network.

**Figure 10 sensors-20-03071-f010:**
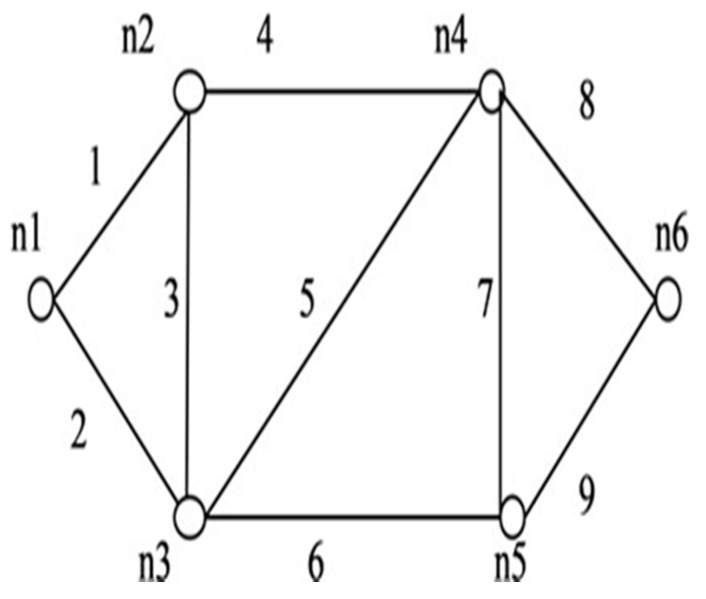
*G* (6, 9).

**Figure 11 sensors-20-03071-f011:**
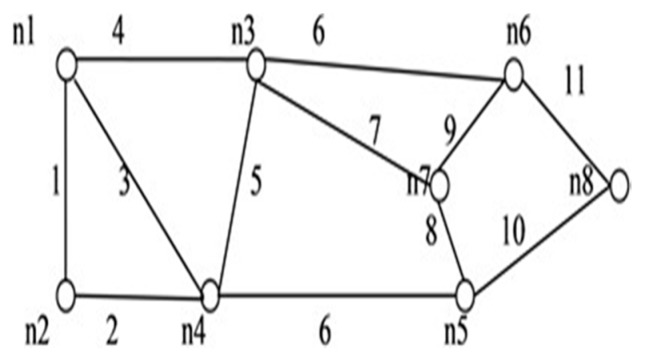
*G* (7, 12).

**Figure 12 sensors-20-03071-f012:**
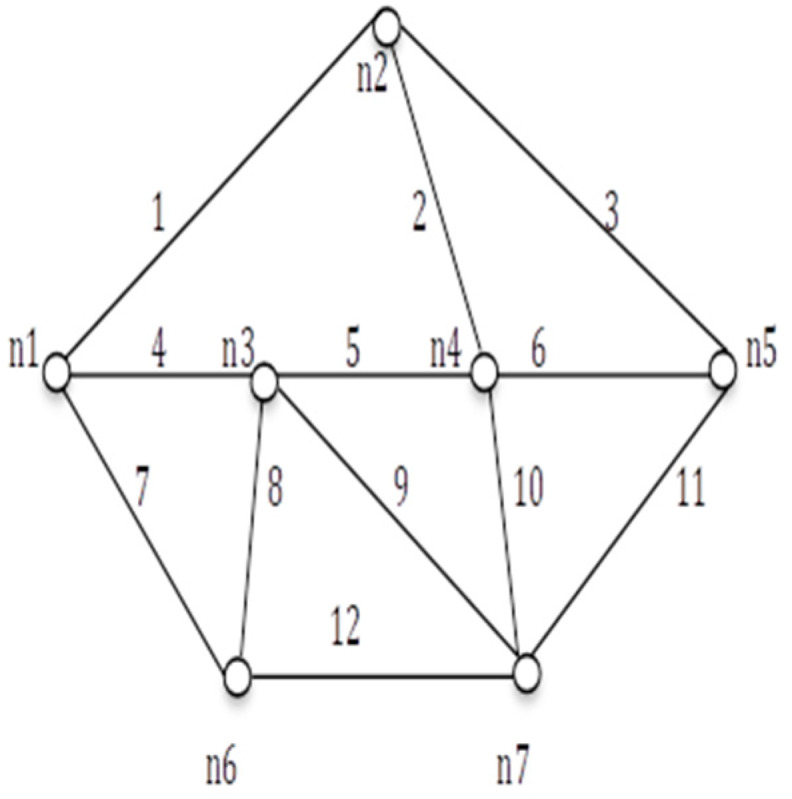
*G* (8, 11).

**Figure 13 sensors-20-03071-f013:**
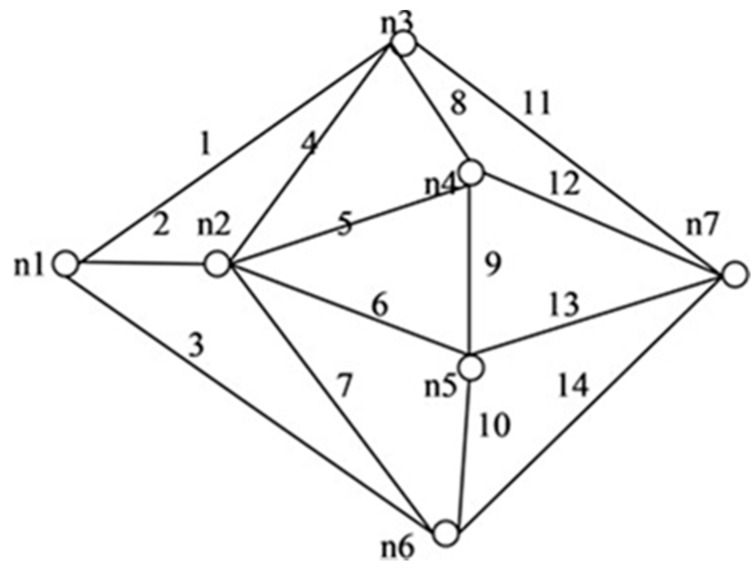
*G* (7, 14).

**Figure 14 sensors-20-03071-f014:**
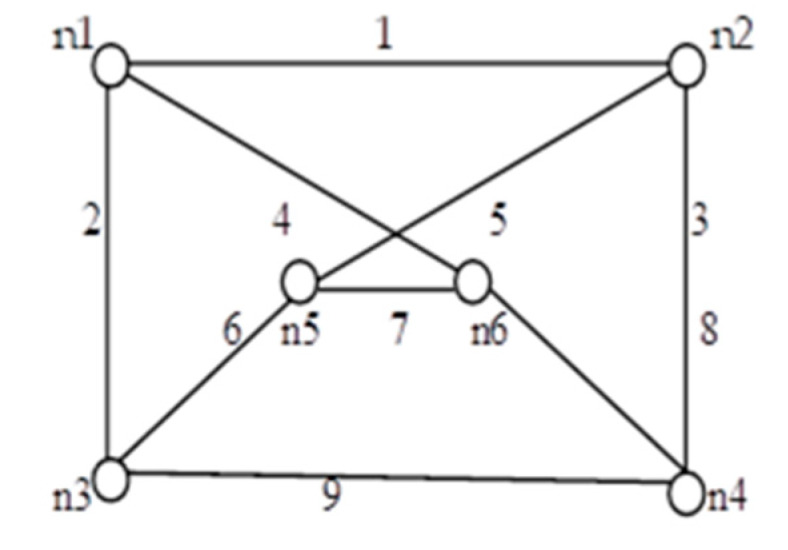
*G* (6, 8).

**Figure 15 sensors-20-03071-f015:**
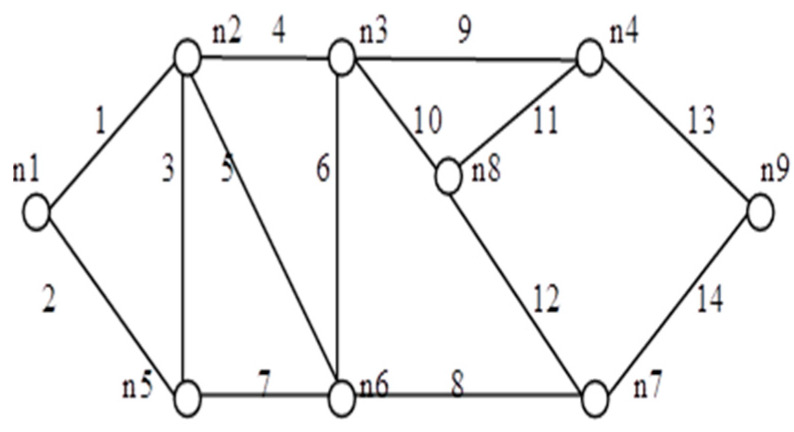
*G* (9, 14).

**Table 1 sensors-20-03071-t001:** Generation of successful spanning trees (STis).

S. No.	Successful Spanning Trees	Appended Edges
1.	1’267	5
2.	12’67	3
3.	126’7	5,4
4.	1267’	3,4
5.	1’2’67	35
6.	1’26’7	45
7.	1’267’	45,35
8.	12’6’7	35,34
9.	12’67’	34
10.	126’7’	34,35,45
11.	1’2’6’7	345
12.	1’2’67’	345
13.	1’26’7’	345

**Table 2 sensors-20-03071-t002:** Disjoint spanning trees (St*i*).

S. No	Spanning Trees
1	1267
2	1’2675
3	12’673
4	126’75
5	126’74
6	1267’3
7	1267’4
8	1’2’6735
9	1’6’2745
10	1’7’2645
11	1’7’2635
12	12’6’735
13	12’6’734
14	12’7’634
15	126’7’35
16	126’7’34
17	126’7’45
18	1’2’6’7345
19	1’2’7’6345
20	1’6’7’2345
21	12’6’7’345

**Table 3 sensors-20-03071-t003:** Comparison of different methods.

Network	Methods	Number of Terms	Number of Multiplications	Number of Additions
Bridge	Proposed			
Network	Disjointing	8	26	7
(Figure 9)	Technique			
	Aggarwal	8	27	7
	Method			
	Satyanarayana	23	65	22
	Method			
Example	Proposed	21	101	20
(Figure 6)	Disjointing			
	Technique			
	Jain and Gopal Method	21	107	20

**Table 4 sensors-20-03071-t004:** Performance and comparison of methods.

Network	Nodes and Edges	Enumeration Time by Proposed Approach (μsec)	No. of Spanning Trees (Disjoint terms)/No. of Path-Sets (Disjoint Terms in Reliability Expression	Reliability by Inclusion, Exclusion Recursive Approach	Reliability by Proposed Approach
Figure 10	*V = 6, E = 9*	3.2 × 10^−3^	81(81)/9(12)	0.61089048	0.99456312
Figure 11	*V = 7, E = 12*	5.8 × 10^−2^	98(98)/6(10)	0.47509544	0.97325916
Figure 12	*V = 8, E = 11*	9.5 × 10^−2^	168(168)/9(11)	0.72987719	0.98689137
Figure 13	*V = 7, E = 15*	2360.63	247(230)/20(30)	0.71898342	0.98292457
Figure 14	*V = 6, E = 9*	3.6 × 10^−6^	80(79)/9(11)	0.6547843	0.9932632
Figure 15	*V = 9, E = 14*	4.0 × 10^−4^	647(644)/44(80)	0.5679345	0.97010434

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
