# Peer review of "Disjoint Spanning Tree Based Reliability Evaluation of Wireless Sensor Network"

_sensors, 2020, doi:10.3390/s20113071_

Round 1
Reviewer 1 Report
This paper is a resubmission of a previous paper. Several aspects of the previous submission have been improved. However, the explanation of the assumptions and the explanation of the results still have to be improved.
The English text quality is poor and still requires an extensive review.
The authors should explain and justify in detail all the assumptions and why they are valid or acceptable.
Why do you assume that "the nodes are perfectly reliable" (line 196)? You should consider some mean time between failures (MTBF) and mean time to repair (MTTR). Maybe the authors should justify that the MTBF is large as compared to packet transmission time and network reconfiguration time, or that only some types of errors are being modelled.
Why do you consider only "two-terminal reliability" (line 230)? The sink node can send data to all sensor nodes (broadcast) or to groups of nodes (multicast), as explained in line 134. Additionally, sensor nodes in a region can self-organize in clusters to provide some common sensor data information to the sink. This has nothing to do with the sink being fixed or mobile as the authors suggest in line 230. Probably, what the authors mean is that they assume the sink is involved in all communications, not that the sink is fixed.
How can "we increase the calculation capacity of the nodes to a desired level" (line 186-187)? The nodes' CPU cannot change.
In table 4, the entry "Figure. 9 Bridge Network" should be below the table top row. Maybe adding "Network" in the first column top row, as a title of that column would improve clarity. Which one is the network of the bottom row, labeled "Example"? Is it the example of figure 3, the example of figure 6, or the example of figure 2? The text in line 556 refers Figure 2, which does not look correct.
Please explain the different columns in table 5. What is the "iIIE Recursive" value? What is the "Proposed Approach" value? What is the "Spanning tree" value? What is meant by "window environment" in line 600? Is it Microsoft Windows environment? You are claiming that table 5 compares the proposed approach with other approaches in the literature, but which are the values of your algorithm and what are the values of the other approaches? How can you claim, in the conclusions section, computational efficiency of the proposed approach based on table 5, since a single CPU time is presented for each network? What about the other algorithms in the literature?
What is the "spatial efficiency of the proposed method" (line 626) claimed to be well explained in Table 5?
Several additional minor corrections are recommended:
- network performance in line 84 is usually related to parameters such as bandwidth, throughput, latency, jitter, error rate, not reliability.
- paper [2] referenced in line 131 is not authored by Colbourn.
- expand acronyms on first use. What is RBD on line 272?
- What is an "Immediate successful spanning tree" (line 305)?
- consider replacing "rest edges" on line 321 and line 422 with "remaining edges" or "rest of the edges".
- Why does rule 1 end with "For the network shown in Figure. 3". Maybe this text should be in the following paragraph.
- Rules 2 and 3 start with quotation marks, but do not end with quotations marks. References to figures probably should be outside the rules.
- replace "note" by "node" in line 471.
- why do you classify other approaches as tedious (e.g. lines 621)? This looks very little scientific.
- avoid contractions in scientific text, like "don't" in line 348 and 509.
- replace "Ramire z" by "Ramirez" in line 602.
- reference [13] has missing information like date, pages and journal number.
- the title of reference [28] is incomplete.
- several references have author initials without a full stop. Some have initials before the surname and other have initials after the surname. Please respect the template.
- the template recommends to include the digital object identifier (DOI) for all references where available.
Reviewer 2 Report
Spanning Tree Protocol (STP) has been included in the IEEE 802.1Q-2014 standard. It is widely used in LANs. The need for the Spanning Tree Protocol (STP) arose because switches in local area networks (LANs) are often interconnected using redundant links to improve resilience should one connection fail. However, this connection configuration creates a switching loop resulting in broadcast radiations and MAC table instability. If redundant links are used to connect switches, then switching loops need to be avoided.
The article deals with the problem of network reliability using the STP protocol. This is an important issue in building a wireless sensor network. The simplified assumptions (192-205) are not related to real networks. Second assumption (196) and assumption (217) and reduction of redundant connections according to IEEE802.1Q creates a trivial statistical problem (conditional probability). Drawing 4 and description do not match (350). The reliability formula was not presented. Which dependencies does the algorithm use (382)? STP provides path mapping (MAC address mapping). How is this included in the algorithm? (Doi: 10.1109 / IEEESTD.2016.7374647) Tables 2 and 3 present the results of the experiment. What did the experiment look like, what solutions and methods of measurement were used? Please describe the experiment, enable it to be reproduced.
In 2001, the IEEE introduced Rapid Spanning Tree Protocol (RSTP) as 802.1w. RSTP provides significantly faster spanning tree convergence after a topology change, introducing new convergence behaviors and bridge port roles to do this. RSTP was designed to be backwards-compatible with standard STP. While STP can take 30 to 50 seconds to respond to a topology change, RSTP is typically able to respond to changes within 3 × Hello times (default: 3 times 2 seconds) or within a few milliseconds of a physical link failure. The Hello time is an important and configurable time interval that is used by RSTP for several purposes; its default value is 2 seconds. Standard IEEE 802.1D-2004 incorporates RSTP and obsoletes the original STP standard.
Round 2
Reviewer 1 Report
This paper has been significantly improved over the previous version and can now be considered for publication.
Only a minor correction is proposed: column "Enumeration Time" in Table 5 is missing the units. From the previous paper version, the units for the "Enumeration time" of the first 4 networks were microseconds. Maybe the "Enumeration time" of the last 2 networks is using different units.
Author Response
Please see the attachment.

This manuscript is a resubmission of an earlier submission. The following is a list of the peer review reports and author responses from that submission.
Round 1
Reviewer 1 Report
This paper is concerned with preserving network connectivity in the presence of link failure(s) and it proposes an algorithm based on spanning trees to do so. The work is not really specific to wireless sensor networks --- in fact the assumption made here that the network is static seems to make this work less appropriate for wireless networks.
I had difficulty understanding the algorithm, which should be presented more formally. Even the central concept of disjoint spanning trees is not defined. Aside from some empirical results on a few small networks, no time- and space-complexity analysis is presented. The algorithm seems to enumerate some spanning trees: how many will be generated? There can be a huge number of them in a large graph. Will the proposed approach scale? No proof of correctness is given either. It is only experimented on very small networks despite stating earlier on that wireless networks may have "in the order of hundreds of nodes".
The authors should avoid unsupported claims that their algorithm is superior and that the others' are "complex and tedious".
The structure of the paper should be reworked: for example Sections 4 and 5 (Terminology, Abbreviations) appear too late in the paper. There are many problems with English syntax. Figure 2 is illegible.
Reviewer 2 Report
The English text quality is poor and requires an extensive review.
Why do you assume that "edge failures being statistically independent" (line 186)? Communication paths involve multiple sequences of edges, so the network links should not be independent of nearby network links.
Why do you assume that "the nodes are perfectly reliable" (line 191)? You should consider some mean time between failures (MTBF) and mean time to repair (MTTR). In equations (1) and (2), you assume nodes may fail, which contradicts the assumption of line 191.
How can you know the reliability of every possible link (line 200)?
Why do you consider only "two-terminal reliability" (line 211)? The sink node can send data to all sensor nodes (broadcast) or to groups of nodes (multicast), as explained in lines 126-128. Additionally, sensor nodes in a region can self-organize in clusters to provide some common sensor data information to the sink.
Equation (2) is not explained. What is pr(n)?
Fig. 2 is of very poor quality and the text unreadable.
What is meant in section 6 by "failed IST" (line 305), "Fail the Edges" (line 316), "Fail two edges" (line 318)? Maybe what is meant is to consider that links have failed and thus, the corresponding network edges should be deleted. But what is a failed Initial Spanning Tree (IST)?
How can r3 fail (line 330) and be a part of the spanning tree (Figure 4)?
Why do you consider in the analysis link failures and not node failures? In a WSN, a link can cause temporary transmission errors. When the node battery is exhausted or some device malfunctions, the entire node fails, not just some link.
In table 2, links with a ' are the failed links? What is "Sr." in the first column? You should explain the table.
Please explain the equations in line 488 and 490. Why is "i" multiplied by (STi)? What is the resulting reliability for the network of the example? How do you determine the probability of each STi?
Table 7 (line 552) only presents CPU times for the proposed algorithm. What about the other algorithms in the literature? Please explain the different columns in table 7.
Several additional minor corrections are suggested:
- network performance in line 76 is usually related to parameters such as bandwidth, throughput, latency, jitter, error rate, not reliability.
- paper [2] referenced in line 125 is not authored by Colbourn.
- line 127, replace "uni cast" by "unicast"
- line 128, replace "multi cast" by "multicast"
- line 130, replace "references [23]" by "reference [23]"
- why do you classify other approaches as tedious (e.g. lines 123, 135, 141) or cumbersome (line 498)? This looks very little scientific.
- avoid contractions in scientific text, like "aren't" in line 172, "don't" in line 408.
- network efficiency (line 192) is not related to reliability.
- on line 218, there should be "m" sensing devices and not "n".
- equations should have a single full stop at the end, not multiple full stops. Equations should be numbered.
- expand acronyms on first use. What is RBD on line 255?
- There is no table 5 and 6. Tables numbers go from 4 to 7.
- references should have author names and initials, not just initials as in references [1,18,19,20,21,23,25,27,33,34,36,40,41,42]
Reviewer 3 Report
The article deals with a very important problem of data transmission reliability in wireless sensor networks. Such networks are increasingly used in industrial environments based on existing standards ( 802.11 (Wi-Fi), 802.15.1 (Bluetooth) czy 802.15.4 (ZigBee, HART) and customized data transfer algorithms (6LoWPAN, Thread, WirelessHART, LoRaWAN, Sigfox). Design and construction of such a network requires calculation of reliability parameters. The article discusses the problem of modeling, calculating spanning trees and assessing the reliability of WSN oriented communications networks.
Section 3.1 presents the assumptions for the calculation. These are very simplified assumptions, I suggest pointing out the differences between these assumptions and the practical implementation of the WSN. The existing data transmission standards use error correction and resumption algorithms. Data buffers are also used to hold the package until a connection is made. The assumptions omitted such mechanisms.
I think that the Sum of Disjoint product (SDP) method for reliability evaluation is a very apt approach. Figure 2 is quite illegible (bad resolution).
The discussion and conclusions are factually correct. The results contribute to the audit of the reliability of technical systems. They can be of great importance in the design of industrial solutions.